# The Influence of Selected Meteorological Factors on the Prevalence and Course of Stroke

**DOI:** 10.3390/medicina57111216

**Published:** 2021-11-08

**Authors:** Katarzyna Zaręba, Anetta Lasek-Bal, Sebastian Student

**Affiliations:** 1Medical University of Silesia, 40-055 Katowice, Poland; kaat.zareba@gmail.com; 2Department of Neurology, School of Health Sciences, Medical University of Silesia, 40-055 Katowice, Poland; 3Faculty of Automatic Control, Electronics and Computer Science, Silesian University of Technology, 44-100 Gliwice, Poland; sebastian.student@polsl.pl; 4Biotechnology Center, Silesian University of Technology, 44-100 Gliwice, Poland

**Keywords:** stroke, weather factors, meteorological parameters, post-stroke functional state

## Abstract

*Background*: The objective of this study was to evaluate the impact of weather factors on stroke parameters. *Methods*: This retrospective study analyzed the records of stroke patients concerning the influence of meteorological conditions and moon phases on stroke parameters. *Results*: The study group consisted of 402 patients aged between 20 and 102; women constituted 49.8% of the subjects. Ischaemic stroke was diagnosed in 90.5% of patients and hemorrhagic stroke was diagnosed in 9.5% of patients. The highest number of hospitalizations due to stroke was observed in January (48 events); the lowest number was observed in July (23 events). There was no statistically significant correlation between the meteorological parameters on the day of onset and the preceding day of stroke and the neurological status (NIHSS) of patients. Mean air temperature on the day of stroke and the day preceding stroke was significantly lower in the group of patients discharged with a very good functional status (≤2 points in modified Rankin scale (mRS)) compared to the patients with a bad functional status (>2 points in mRS); respectively: 7.98 ± 8.01 vs. 9.63 ± 7.78; *p* = 0.041 and 8.13 ± 7.72 vs. 9.70 ± 7.50; *p* = 0.048). Humidity above 75% on the day of stroke was found to be a factor for excellent functional state (RR 1.61; *p* = 0.016). The total anterior circulation infarcts (in comparison with stroke in the other localization) were more frequent (70%) during a third quarter moon (*p* = 0.011). The following parameters had a significant influence on the number of stroke cases in relation to autumn having the lowest number of onsets: mean temperature (OR 1.019 95% CI 1.014–1.024, *p* < 0.000), humidity (OR 1.028, CI 1.023–1.034, *p* < 0.0001), wind speed (OR 0.923, 95% CI 0.909–0.937, *p* < 0.0001), insolation (OR 0.885, 95% CI 0.869–0.902, *p* < 0.0001), precipitation (OR 0.914, 95% CI 0.884–0.946, *p* < 0.0001). *Conclusion*: Air humidity and air temperature on the day of stroke onset as well as air temperature on the day preceding stroke are important for the functional status of patients in the acute disease period. A combination of the following meteorological parameters: lowered mean temperature and low sunshine, high humidity and high wind speed all increase the risk of stroke during the winter period. High humidity combined with high precipitation, low wind speed and low sunshine in the autumn period are associated with the lowest stroke incidence risk. A possible relationship between phases of the moon and the incidence requires further investigation.

## 1. Introduction

Fluctuations in weather factors can unsettle the homeostasis of the human body and promote the onset and/or exacerbation of nervous system diseases. By using epidemiological data, attempts are undertaken to determine the trend in annual stroke incidence. Ischaemic stroke cases have been observed to increase during the winter or summer months; however, some investigators failed to observe any seasonal variability in stroke incidence. Most studies have shown decreased hospitalization rates due to ischaemic stroke in the summer months [1,2,3]. Similarly, the incidence of hemorrhagic stroke shows a seasonal trend, with the highest incidence in the winter months [4]. Ambient temperature is a meteorological parameter most frequently identified as a stroke incidence factor; its low value has been shown to be a predisposing factor for stroke [2,3]. Attention is given not only to the effect of absolute values of air temperature on acute cardiovascular incidents and mortality but also to the importance of diurnal differences in temperature. To date, the effects of atmospheric pressure, precipitation, sunshine, wind speed and humidity on stroke incidence have also been studied. The results of studies evaluating the influence of individual meteorological factors on stroke incidence are inconclusive and often contradictory. The differences might be due to the fact that those studies were conducted in various geographical regions and different climatic zones. Local weather conditions (a specific microclimate) affecting the population inhabiting a certain area are also of some importance; this may partly explain why conclusions differ from author to author. The assessment of the impact of weather conditions on stroke incidence parameters, the course of stroke and stroke-related mortality is important in the context of climate change, with global warming occurring over the previous decade. According to the data published by the World Health Organisation (WHO), 12.6 million deaths (23% of all deaths worldwide) result from modifiable environmental factors such as those related to climate [5]. According to the 2015 Lancet Commission, managing the effects of climate on the human body in the 21st century may be the best opportunity to improve the health of the global population in relation to cardiovascular disease, including stroke [6].

The main objective of this study was to evaluate the impact of selected meteorological factors on the incidence and type of stroke, and on the course and prognosis during acute stroke period.

## 2. Materials and Methods

This retrospective study analyzed the case records of stroke patients of the Department of Neurology in Katowice hospitalized due to stroke between 1 January and 31 December 2015. The main criterion was stroke (ICD-10: I63, I64, I60, I61) as diagnosed in accordance with the definition provided by the WHO and confirmed by neuroimaging investigations (CT and/or MRI of the head). The analysis covered patients whose onset of clinical symptoms occurred within 24 h prior to admission. Only patients hospitalized ≤14 day were included in this study (the patients’ functional status was examined on discharge between 10–14 days from stroke-onset). The patients with transient ischemic attack (TIA) or traumatic subarachnoid hemorrhage were not eligible for the study. Stroke severity was assessed on the first day of disease (at baseline) using the NIHSS (National Institutes of Health Stroke Scale) and the functional status was assessed on 14th day of stroke in accordance the modified Rankin scale.

The case history of each patient regarding stroke risk factors such as age, arterial hypertension, lipid disorders, ischaemic heart disease, (past) acute coronary syndrome, heart failure, atrial fibrillation, diabetes, history of stroke and hemodynamically significant stenosis of internal carotid arteries as well as the method of ultra-acute phase treatment was collected.

AF (paroxysmal, persistent, chronic) was diagnosed on the basis of previous (pre-stroke) patient medical records or ECG, or 24-h ECG monitoring performed during stroke-related hospitalization (≤14 days after onset). The diagnosis of AH was consistent with the recommendations of the European Society of Cardiology (ESC); DM was diagnosed according to the criteria of the Diabetes Association; dyslipidemia was defined according to the ESC recommendations (Guidelines for the Management of Dyslipidemias). The degree of the common carotid artery stenosis and/or internal carotid artery stenosis was assessed according to the NASCET criteria.

Additionally, stroke risk factors investigated as part of causal diagnosis during hospitalization and after discharge were included in the assessment. Clinically significant stenosis of the internal carotid artery was defined as ≥50% stenosis visualized by Doppler ultrasound or angio-CT of the head. In each patient with no AF in the basic electrocardiogram recording, a 24-h electrocardiography device monitoring was performed during hospitalization. AF was diagnosed based on previous patient records (pre-stroke period) or ECG, or 24-h ECG monitoring. Similarly, each patient had a transthoracic echocardiogram performed; those below 55 years of age underwent a transesophageal echocardiogram. The parameters: arterial blood pressure and heart rate and peripheral blood count (hematocrit, hemoglobin concentration, thrombocyte and red blood cell count) we collected at admission. The functional status of patients was assessed on the 14th day of stroke as per the modified Rankin scale. Clinical cases were analyzed regarding sex, the age at stroke onset, stroke risk factors, the location of stroke focus in accordance with the OSCP classification, the phenotypic classification of stroke as per ASCOD scale. 

Weather data were obtained from the website recommended by the Institute of Meteorology and Water Management (ogimet.com, accessed on 30 June 2019). The meteorological parameters obtained between 1 January and 31 December 2015 were acquired from the meteorological station in Katowice (50°14′ N, 19°02′ E) located at 284 m above sea level. The following meteorological data were collected: air temperature (maximum, minimum and mean daily temperature) expressed in degrees Celsius, relative humidity expressed in percentage, sunshine as the total time of direct sunlight expressed in hours per day, atmospheric pressure expressed in hPa, the daily value was averaged using hourly measurements, wind direction and wind speed expressed in km/h, precipitation expressed in millimeters and finally, the differences between temperatures and atmospheric pressure values on the day preceding stroke and the day of stroke. Moon phase data were obtained from the timeanddate.com website (accessed on 30 June 2019) for the Katowice area between 1 January and 31 December 2015. 

The above clinical and meteorological parameters were analyzed to assess the relationship between: (1) the meteorological conditions on particular days of the year and the number of strokes, stroke type, neurological status of patients in accordance with the NIHSS on the first day of stroke; (2) the meteorological conditions on the first day of stroke and the functional status of patients on their last day of hospitalization; (3) the meteorological conditions and the parameters related to the circulatory system (blood pressure, heart rate) and peripheral blood cell count (hemoglobin, hematocrit, thrombocyte and red blood cell count) at hospital admission; (4) the variability of temperature and arterial blood pressure at 24 h before stroke onset and the number of strokes; (5) the number and type of strokes during specific lunar phases: full moon, new moon, first and third quarter moon.

The study received a positive opinion from the Bioethics Committee at the Medical University of Silesia in (KNW/0022/KB/285/18).

### Statistical Analysis

Parametric variables were characterized using mean, standard deviation, median, minimum and maximum values. Nonparametric variables were described by numbers and percentages represented in the study group. Comparisons between groups utilized the following methods for parametric variables: Student’s T-test for separable variables in the case of variables with normal distribution; Mann–Whitney U test for those variables which failed to meet the criteria for normal distribution. Nonparametric variables were compared using the Chi-squared test. Logistic regression was used for univariate/multivariate analysis by determining the relative risk of the effect of a specific variable on an independent variable. Regression analysis was performed to assess the effect of the following parameters on the patients’ good functional status (≤2 points in modified Rankin scale (mRS)) on discharge day (between 10–14 after onset): age, gender, diabetes mellitus, arterial hypertension, atmospheric pressure > 985 hPa, humidity > 75%, wind speed > 8 m/s, insolation > 3.6 h, precipitation > 0 mm, rising moon. Statistical analyses were performed using R ver. 3.6.1. 

Poisson regression was applied to the data regarding the count rates of stroke throughout one year. R environment (ver. 3.6.1) with stats and sandwich package were also used. The model meets the requirements of linearity, is well accommodated to the count rates data, and the data are distributed evenly. The method of robust estimator proposed by Cameron and Trivedi [7] was used to determine the standard error for the parameter estimates. The model is statistically significant and meets the goodness-of-fit criterion.

## 3. Results

The study group consisted of 402 patients aged between 20 and 102; women constituted 49.8% of the subjects. Ischaemic stroke was diagnosed in 90.5% of patients and hemorrhagic stroke was diagnosed in 9.5% of patients. In 226 (53.8%) patients stroke was localized in the anterior circulation (internal carotid artery supply), in most of them (197, 74.06%) in the left hemisphere. 124 (30.84%) patients were treated with intravenous thrombolysis. In 11 of them, we observed complication rt-PA- related including the symptomatic intracranial bleeding in 3 patients (2.4%). Table 1 presents the clinical features of all subjects along with the comparison of the parameters for patients divided by sex and age. The mean age of women was significantly older than that of men; women more often presented ischaemic heart disease and a worse functional status on the day of discharge. Among women, carotid artery stenosis, stroke associated with large artery disease and stroke located within the posterior cerebral circulation were significantly less frequent. Older individuals were significantly more often burdened with cardiovascular disease as following: diabetes mellitus, ischaemic heart disease, arterial hypertension, atrial fibrillation. The cardioembolic stroke was significantly more often in patients older than 65 year in comparison with younger. (Table 1). 

The highest percentage of hospitalizations due to stroke was observed in January; the lowest percentage was observed in July (Figure 1).

For all twelve months, PACI was most common among ischaemic strokes (it accounted for 40.5–75% of all strokes); TACI was the rarest (≤7.7%). As per ASCOD classification, cardioembolic stroke and classified to the category “other” were most often types of stroke in monthly analyses. In heterogenic category (other), the most common were three strokes as follows: cancer-related stroke (34, 34% of all in this category), migrainous stroke (15, 15%) and stroke in the course of antiphospholipid syndrome (11, 11%).

The annual distribution of the mean values of meteorological parameters is presented in Figure 2. (Figure 2). There was no statistically significant correlation between the assessed meteorological parameters on the day of onset and the preceding day and the neurological status of patients on the first day of stroke. (Appendix A in Supplement) Atrial fibrillation (HR 1.94 *p* = 0.047) was an independent risk factor for severe neurological status (NIHSS 13–42) among the studied clinical and meteorological parameters in whole study group as well as among the patients with ischaemic stroke (HR 2.06 *p* = 0.039). None of the meteorological parameters proved to be statistically significant factor for the neurological status as per the NIHSS.

The mean temperature on the day of stroke and on the day preceding stroke were important for the functional state of the patients on the day of discharge. Table 2. The following favorable prognostic factors (mRankin ≤ 2) were identified: male gender, age below 65 years and no atrial fibrillation, humidity on the day of stroke above 75%. (Table 3).

There were no significant differences regarding the meteorological parameters on the day of stroke between the patients with ischaemic and hemorrhagic stroke (Appendix A in Supplement).

No statistically significant differences were found in relation to meteorological parameters on the day of ischaemic stroke onset between the different age groups of patients.

There was no significant effect of daily differences in air temperature or atmospheric pressure on the number of hospitalizations (>3 or ≤3) due to stroke between the day preceding stroke and the day of stroke.

The mean value of systolic arterial blood pressure measured in patients on the day of admission was significantly higher on the days of moderate or weak wind (≤8 m/s) than on the days of strong wind (>8 m/s). (Table 4).

No statistically significant differences were observed in selected blood counts or clinical parameters on the days of lower (<10 °C) and higher (>15 °C) air temperature, on hot days (maximum temperature > 25 °C) and on frosty days (minimum temperature < −10 °C), depending on precipitation, on dry days (relative air humidity ≤ 70%) and on days of moderate or high air humidity (>70%), depending on atmospheric pressure changes or sunshine (on days of low (<1 h) and high (>4 h) sunshine).

The following meteorological parameters demonstrate a significant influence on the increase in the number of strokes in winter (a season of the highest incidence): lower average temperature and insolation, high humidity and wind speed. (Table 5, Figure 3).

The following meteorological parameters demonstrate a significant influence on the decrease in the number of strokes in autumn (a season of the lowest incidence): high humidity, high precipitation, low wind speed and low sunshine. (Table 5, Figure 3).

The meteorological parameters analyzed, except atmospheric pressure, showed a significant influence on the number of stroke cases in individual seasons (in relation to autumn having the lowest number of onsets).

The highest number of hospitalizations due to stroke occurred during a first quarter moon (113). The fewest strokes occurred in the period of full moon (88). 

There was no statistically significant difference in the incidence of hemorrhagic strokes or ischaemic strokes during individual moon phases. The lowest number of hospitalizations due to intracranial hemorrhage occurred in the new moon phase (18.4%); the lowest number of hospitalizations due to ischaemic stroke occurred in the full moon phase (21.2%). (Table 6).

It was statistically significant that the total anterior circulation infarcts (TACI) were more frequent during the third quarter moon phase (70%). No significant influence of moon phases on other types of stroke was observed.

## 4. Discussion

Stroke is a polyetiological disease. Genetic, individual, clinical and environmental factors are considered as the underlying disease factors. Considering the importance of biorhythms for homeostasis in various organisms and the integrity of biological processes in nature, attention is paid to the potential influence of meteorological conditions on the incidence and course of stroke. However, the results of previous studies in this area are inconsistent; to date, they have failed to provide us with information about the significance and degree of the effect of weather conditions on stroke incidence and the course of the acute stroke phase [8,9,10,11,12,13,14,15,16,17,18]. Attempting to define and organize the meteorological factors of potential importance for stroke incidence, the course of the acute phase and the early post-stroke functional status of patients as well as a keen interest in this topic were all the reasons for starting the investigations as part of this study. The results of previous studies that focused on the association between the weather, stroke incidence and stroke seasonality to a large extent, suggest an increased incidence of stroke during the winter period, which is also consistent with the results of this study. Most hospitalizations for acute stroke occurred in winter; the lowest number occurred during the months of autumn. The winter period is noted for lower mean air temperatures, which—according to the results of this study and those by other authors—increase the risk of stroke. Similar results concerning the seasonality of stroke were also obtained by other authors. [2,3,19]. On the other hand, in the Arab population, a significant increase in ischaemic strokes was observed in the summer, with a marked decrease in the winter months [10]. In a study covering the population of Zagreb (Croatia), an increased number of ischaemic strokes was reported in the spring period; hemorrhagic strokes were reported in winter [20]. However, a study conducted in South Korea found no seasonality factors tied to ischaemic stroke incidence. Ref. [21] Circannual biological rhythms are determined by the variability in diurnal ambient temperature and the length of days and nights; therefore, their effects on the human body may depend on latitude and vary globally from region to region [22]. The rather consistent results obtained by European investigators and quoted above can support this hypothesis.

Ambient temperature is considered the main meteorological factor associated with the seasonality of stroke [2]. Many authors have shown a negative correlation between high temperature and the number of both hemorrhagic and ischaemic strokes [8,9,23,24]. The results of this study did not show any statistically significant relationship between the number of certain types of stroke and air temperature. Most probably, the analysis of the influence of a single meteorological parameter is of less importance than the influence of several weather factors acting simultaneously on the human body. The maximum wind speed some days before stroke and change of atmospheric pressure in the last 24 h were found to increase the cases of ischemic stroke [25].For patients residing in the Arabian Peninsula, where mean annual temperatures and daily sun exposure are high, a positive correlation was observed between stroke incidence and these parameters [10]. A study by Xu et al. showed an association between low temperature and increased arterial blood pressure, which may indirectly alter the incidence of cardiovascular diseases and explain the mechanism responsible for the effect of low temperature on stroke incidence [26]. Su et al. [27] demonstrated increased activity of the renin-angiotensin system under the influence of low temperature and imbalances between Matrix Metalloproteinase-9 and its inhibitor, TIMP-1. MMP-9 is present in blood vessel walls and is responsible for the degradation of the extracellular matrix. This is why changes in the activity of this enzyme in atherosclerotic vessels may result in an acute vascular incident [27]. The results of our study do not confirm a relationship between air temperature, other meteorological factors and hemoglobin levels, hematocrit, erythrocyte and platelet counts in patients on the first day of stroke. It should be stressed that our study showed the effect of temperature on the functional status of patients after the acute phase of stroke. A lower mean air temperature (range: −7.2–27 °C) on the day of onset and on the preceding day was associated with a better prognosis regarding the functional status of patients at discharge. There are some reports on the negative influence of diurnal variations in temperature and atmospheric pressure (and not of absolute values of these parameters) on stroke risk during wintertime [28]. Temperature changes probably affect systemic arterial pressure values, lipid metabolism, inflammatory response, immune system and fibrinogen concentration, which in turn affects the cardiovascular system and thrombus formation [29,30,31]. Large diurnal temperature fluctuations increase cellular oxygen uptake, catecholamine release, arterial pressure and blood viscosity [31,32]. According to some investigators, a cerebrovascular incident occurs 2–3 days after the daily temperature fluctuations [31]. In this study, a shorter time interval was analyzed, and no significant association between these parameters was found.

Fluctuations in atmospheric pressure can promote arterial blood pressure instability and hemodynamic changes in the circulatory system: both the systemic and cerebral systems [9]. In the presented study, there was no increase in the number of admissions to the stroke unit (≥3) on days of large atmospheric pressure fluctuations. Jimenez-Conde et al. demonstrated an association between stroke incidence and atmospheric pressure variability at 24 h before onset. In a study by Gunes et al., atmospheric pressure variability increased the risk of ischaemic stroke [25]. Similar results have been obtained by other authors. [10,28]. It has been suggested that high atmospheric pressure acts as an external stress factor to compromise the stability of atherosclerotic plaque [33]. Atmospheric pressure instability can also modify the secretory activity of the endothelium [16,34].

The presented study analyzed the potential effect of air humidity on the incidence of stroke and its course; it showed that relative humidity above 75% is a favorable prognostic factor for the post-stroke functional status of patients. However, no relationship was shown between air humidity and blood count parameters. Knezovic et al. observed a negative correlation between stroke rates and relative air humidity [20]. In contrast, studies conducted in Paris showed that higher air humidity combined with exposure to air pollution increased the risk of stroke [35]. This may be caused by increased transmission of dust in humid air and an increased exposure to toxic substances. Dry air, especially in high temperatures, increases evaporation and water loss, which—along with inadequate hydration—can promote blood thickening and blood viscosity increase. The results of the analysis regarding the influence of several meteorological parameters on stroke incidence are worth stressing. High humidity with high wind speed at a reduced mean temperature and lower sunshine significantly raised the increase in stroke rate for winter. However, high humidity combined with high precipitation, low wind speed and low sunshine in the autumn period reduced stroke incidence. As per the results published by Matsumaru et al., a radical change in humidity is of more importance for stroke risk than the value of absolute humidity [2].

There are several reports regarding wind speed as a potential factor influencing stroke incidence. Kim et al. showed an increased risk of ischaemic stroke on days of higher wind speeds [36]. A similar relationship in a group of older patients was observed by Tamasauskiene et al. [24]. The presented study shows a significant influence of wind on the blood pressure of patients hospitalized due to stroke. The value of systolic arterial pressure measured in patients on the day of admission was significantly higher on the days of moderate or weak wind (≤8 m/s). Wind speed in combination with temperature, sunshine and humidity played a significant role in stroke rates during the highest and lowest incidence seasons, i.e., in winter and autumn, respectively.

Although the instability of external environment caused by changing weather conditions can upset the homeostasis of living creatures, our study did not demonstrate a significant influence of variable meteorological phenomena on the incidence of stroke. Variables of air temperature and atmospheric pressure on the day preceding stroke onset were evaluated along with the effect of several co-occurring weather parameters on the number of hospitalizations due to stroke. Gunes et al. evaluated the influence of weather changes on the incidence of ischaemic stroke; they demonstrated an association between a change in atmospheric pressure at 24 h before onset and an increased ischaemic stroke rate [25]. Lim et al. demonstrated an association between diurnal temperature range, arterial pressure, temperature variability within 24 h preceding stroke and the incidence of ischaemic stroke cases [21]. Rakers et al. suggest an increased risk of stroke due to rapid changes in the following meteorological parameters: temperature, atmospheric pressure and air humidity in temperate climates, particularly in patients with increased cardiovascular risk [37].

To date, not many investigations focused on the potential association between moon phases and stroke incidence have been conducted. In this study, it was found that a significantly higher incidence of TACIs occurred during the third quarter moon phase. The conclusions reported by other authors are inconsistent [38,39,40].

It seems that the effect of weather conditions on the cardiovascular system is related to many aspects. Air pollution from transport, household heating systems and industry is of great significance. The distribution of air pollution particles depends on humidity, temperature and sunshine [41]. The above-mentioned parameters additionally influence stroke prevalence parameters in populations inhabiting urban areas. An assessment regarding the importance of meteorological parameters in relation to the incidence and course of stroke will make it easier to develop healthcare strategies, which would reduce the impact of exogenous factors on the human body and allow us to effectively manage the risk of stroke both for individuals and societies as a whole.

The strength of the manuscript is a comprehensive analysis of the influence of meteorological factors on the occurrence of stroke, as well as the analysis of the relationship between meteorological factors and selected clinical factors and blood count.

The limitations of the presented study: the analysis included patient data gathered from a single institution specialized in stroke treatment; no assessment concerning air pollution levels was conducted (pollutants may modify the effects of meteorological parameters on patients). The next limitation is lack of analysis of the potential correlations between the meteorological parameters and the lifestyle risks of stroke. The other limitation can be the lack the analysis of the influence of meteorological parameters on functional status of patients on 30th day after stroke, but we were unable to collect complete data.

## 5. Conclusions

Air humidity and air temperature on the day of stroke onset, as well as air temperature, on the day preceding stroke are important for the functional status of patients in the acute disease period.A combination of the following meteorological parameters: lowered mean temperature and low sunshine, high humidity and high wind speed all increase the risk of stroke during the winter period.High humidity combined with high precipitation, low wind speed and low sunshine in the autumn period are associated with the lowest stroke incidence risk.A possible relationship between phases of the moon and the incidence of requires further investigation.

## Figures and Tables

**Figure 1 medicina-57-01216-f001:**
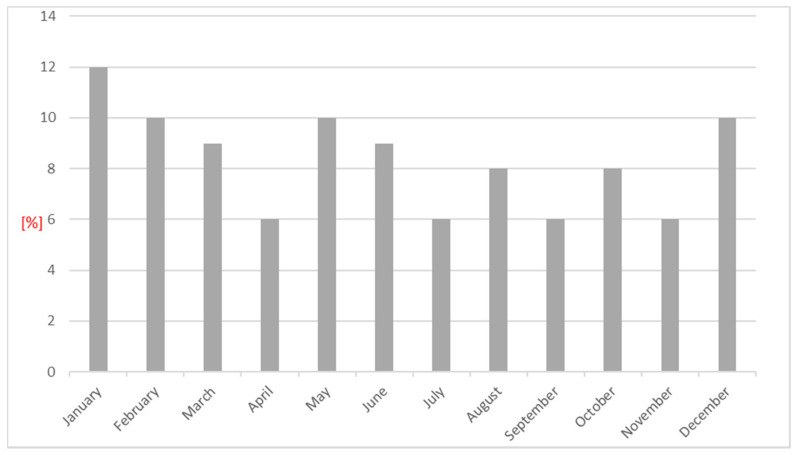
The percentage of patients hospitalized due to stroke by month.

**Figure 2 medicina-57-01216-f002:**
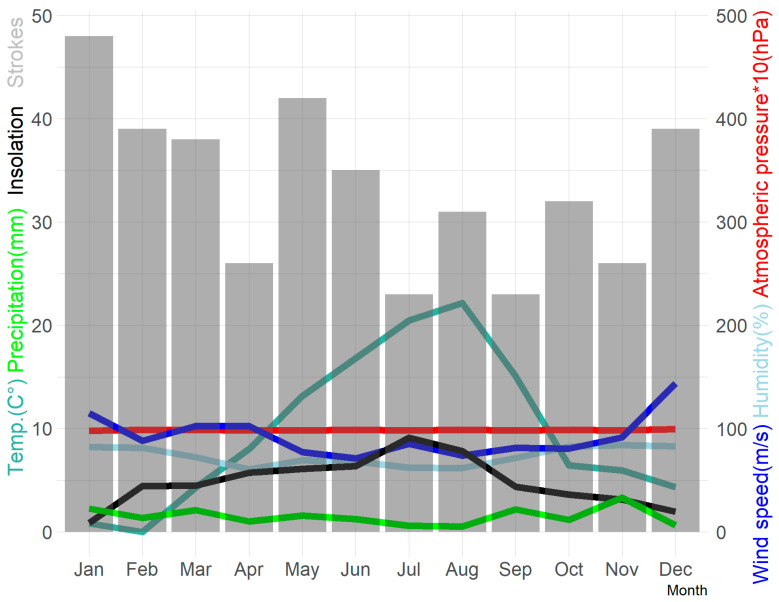
Meteorological parameters and stroke incidence by each month.

**Figure 3 medicina-57-01216-f003:**
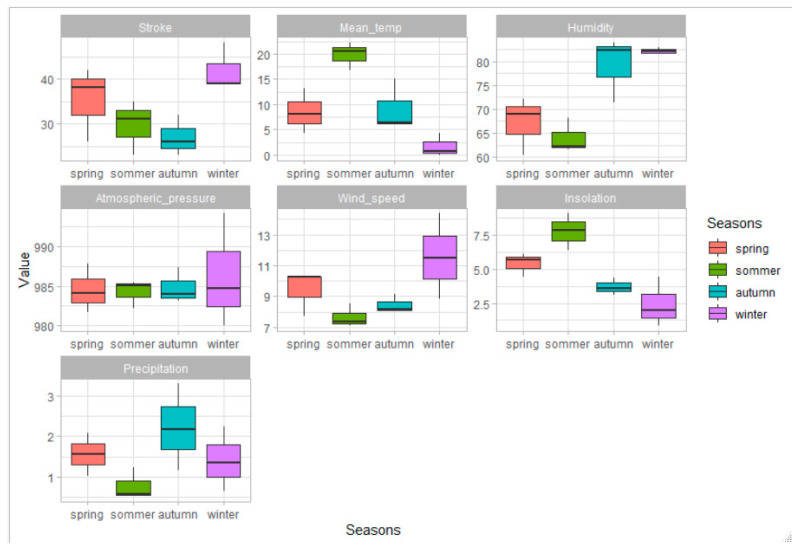
Box plot graph presenting the distribution of all the meteorological parameters for individual seasons. Temp-temperature.

**Table 1 medicina-57-01216-t001:** Demographic and clinical features of patients.

Parameter	*N* = 402	Women *N* = 200	Men *N* = 202	*p*	Patients ≤65 Years of Age *N* = 124	Patients >65 Years of Age *N* = 278	*p*
Age	71.1 ± 13.1 Median 72; range (20–102)	74.7 ± 13.4 Median 76 range (24–102)	67.5 ± 11.9 Median 68 range (20–96)	0.000	-	-	-
Diabetes	119 (29.6%)	63 (31.5%)	56 (27.7%)	0.407	25 (20.2%)	94 (33.8%)	0.006
Ischaemic heart disease	180 (44.8%)	102 (51.0%)	78 (38.6%)	0.013	31 (25%)	149 (53.6%)	0.000
History of myocardial infarction	72 (17.9%)	38 (19.0%)	34 (16.8%)	0.571	20 (16.1%)	52 (18.7%)	0.534
Carotid artery stenosis ≥50% *	72 (17.9%)	27 (13.5%)	45 (22.3%)	0.022	23 (18.5%)	49 (17.6%)	0.824
Lipid disorders	159 (39.6%)	86 (43.0%)	73 (36.1%)	0.160	51 (41.1%)	108 (38.8%)	0.666
Arterial hypertension	353 (87.8%)	177 (88.5%)	176 (87.1%)	0.674	96 (77.4%)	257 (92.4%)	0.000
Atrial fibrillation	123 (30.6%)	65 (32.5%)	58 (28.7%)	0.410	15 (12.1%)	108 (38.8%)	0.000
Previous stroke	103 (25.6%)	54 (27.0%)	49 (24.3%)	0.529	27 (21.8%)	76 (27.3%)	0.238
NIHSS ** mean ± SD; med. (range); IQR (range Q1 Q3)	4.90 ± 5.13 3 (0–30) 3 (1–16)	5.21 ± 5.24 3 (0–23) 3 (0–18)	4.59 ± 5.0 3 (0–30) 3 (0–17)	0.268	4.46 ± 5.10 3 (0–30) 3 (0–21)	5.09 ± 5.14 3 (0–28) 3 (1–26)	0.129
mRS *** mean ± SD; med. (range); IQR (range Q1 Q3)	2.81 ± 1.96 3 (0–6) 3 (0–6); 3(2–5)	3.1 ± 1.96 3 (0–6); 3 (0–6); 3 (2–5)	2.52 ± 1.93 2 (0–6); 2 (0–6); 2 (2–5)	0.004	2.31 ± 2.02 2 (0–6); 2 (0–6); 2 (2–5)	3.04 ± 1.90 3 (0–6); 3 (0–6); 3 (2–5)	0.001
Hemorrhagic stroke	38 (9.5%)	23 (11.5%)	15 (7.4%)	0.163	13 (10.5%)	25 (9.0%)	0.637
Ischaemic stroke	364 (90.5%)	177 (88.5%)	187 (92.6%)	0.163	111 (89.5%)	253 (91.0%)	0.637
LACI	51 (12.7%)	24 (12%)	27 (13.4%)	0.681	21 (16.9%)	30 (10.8%)	0.087
PACI	212 (52.7%)	112 (56%)	100 (49.5%)	0.192	64 (51.6%)	148 (53.2%)	0.763
POCI	91 (22.6%)	37 (18.5%)	54 (26.7%)	0.049	21 (16.9%)	70 (25.2%)	0.068
TACI	10 (2.5%)	4 (2%)	6 (3.0%)	0.532	5 (4.0%)	5 (1.8%)	0.184
ASCOD—A	60 (14.9%)	22 (11%)	38 (18.8%)	0.028	21 (16.9%)	39 (14%)	0.450
ASCOD—S	59 (14.7%)	28 (14%)	31 (15.3%)	0.703	23 (18.5%)	36 (12.9%)	0.143
ASCOD—C	145 (36.1%)	74 (37%)	71 (35.1%)	0.699	23 (18.5%)	122 (43.9%)	0.000
ASCOD—O	100 (24.9%)	53 (26.5%)	47 (23.3%)	0.454	44 (35.5%)	56 (20.1%)	0.001
ASCOD—D	0 (0%)	0	0	0	0	0	0

* ipsilaterally to stroke; ** NIHSS score at admission; *** mRS—modified Rankin scale at discharge; NIHSS—National Institute Health Stroke Scale, LACI—lacunar cerebral ischemia, PACI—partial anterior circulation infarcts, POCI—posterior circulation infarct, TACI—total anterior circulation infarct; ASCOD—phenotypic stroke classification, ASCOD-A—atherosclerosis, ASCOD-S—small vessel disease, ASCOD-C—cardioembolic, ASCOD-O—other, ASCOD-D—dissection.

**Table 2 medicina-57-01216-t002:** Meteorological parameters and functional status of patients as per modified Rankin scale (mRS) on the day of discharge from the Neurology Department.

Parameter	mRS 0–2 *N* = 177	mRS 3–6 *N* = 225	*p*
Mean temperature on the day of stroke [°C]	7.98 ± 8.01 Median 6.1 (−7.2–27)	9.63 ± 7.78 Median 8.8 (−7.2–26)	0.041
Mean temperature on the day preceding stroke [°C]	8.13 ± 7.72 Median 6.9 (−7.2–27)	9.70 ± 7.50 Median 9.3 (−7.2–26)	0.048
Mean atmospheric pressure on the day of stroke [hPa]	984.3 ± 9.2 Median 984.8 (946.9–1004.7)	985.6 ± 7.8 Median 986.9 (946.9–1001.6)	0.129
Mean atmospheric pressure on the day preceding stroke [hPa]	984.5 ± 8.0 Median 984.5 (958.4–1004.7)	985.7 ± 7.5 Median 985.6 (963.5–1004.4)	0.122
Mean relative humidity [%]	75.1 ± 12.9 Median 76.2 (43.2–96.4)	73.2 ± 13.4 Median 72.5 (43.6–95.8)	0.146
Mean wind speed [km/h]	9.8 ± 5.1 Median 8.3 (1.8–24.1)	9.1 ± 4.6 Median 7.8 (1.8–22.2)	0.257

mRS- modified Rankin scale at discharge; Mean air temperature (range: −7.2–27 °C) on the day of stroke and the day preceding stroke was significantly lower in the group of patients discharged from hospital in a better functional status (mRankin 0–2) than in those discharged in a worse functional status.

**Table 3 medicina-57-01216-t003:** Factors with a potential impact on the functional status (≤2 mRS) on the day of discharge from the Neurology Department.

Parameter	*p*	OR
Sex (men)	0.001	2.00
Age ≤ 65 years	0.000	2.28
No atrial fibrillation	0.014	1.72
Humidity > 75%	0.016	1.61

**Table 4 medicina-57-01216-t004:** Mean values of patients’ selected blood counts, vital signs, neurological and functional state in relation to wind speed on the day of stroke onset.

Parameter	Wind ≤ 8 m/s *N* = 202	Wind > 8 m/s *N* = 200	*p*
Ht [%]	39.1 ± 4.8 Median 39.55 Range (23.2–48.4)	39.8 ± 4.6 Median 40.45 Range (25–52.6)	0.550
RBC [M/µL]	4.44 ± 0.57 Median 4.455 Range (2.54–5.88)	4.51 ± 0.57 Median 4.585 Range (2.66–6.11)	0.197
PLTs [K/µL]	235.3 ± 80.7 Median 219 Range (100–578)	230.4 ± 79.3 Median 216.5 Range (73–606)	0.537
Systolic blood pressure [mmHg]	160.2 ± 29.8 Median 160 Range (90–240)	153.7 ± 26.0 Median 150 Range (110–240)	0.021
Diastolic blood pressure [mmHg]	85.7 ± 16.3 Median 80 Range (50–160)	85.6 ± 14.3 Median 80 Range (40–130)	0.973
NIHSS * mean ± SD; med. [range]; IQR [range Q1 Q3]	4.86 ± 4.62 3 (0–22) 3 (0–16)	4.94 ± 5.60 3 (0–30) 3 (0–19)	0.335
mRS ** mean ± SD; med. [range]; IQR [range Q1 Q3]	2.91 ± 1.93 3 (0–6); 3 (2–5)	2.72 ± 2.00 3 (0–6); 3 (2–5)	0.323

Ht- hematocrite, RBC- red blood cell, PLTs- platelets; * NIHSS (National Institute Health Stroke Scale) score at admission; ** mRS- modified Rankin scale at discharge.

**Table 5 medicina-57-01216-t005:** Poisson regression model regarding the number of stroke cases in each season depending on weather conditions (AIC = 84.24).

Parameter	Risk Ratio	Estimate	Robust SE	Pr (>|z|)	CI 95%	
					LL	UL
Mean temperature [°C]	1.019	0.019	0.003	<0.0001	0.014	0.024
Relative humidity [%]	1.028	0.028	0.003	<0.0001	0.023	0.033
Atmospheric pressure [hPa]	0.997	−0.003	0.004	0.37	−0.01	0.004
Wind speed [km/h]	0.923	−0.08	0.008	<0.0001	−0.096	−0.065
Insolation [h]	0.885	−0.122	0.01	<0.0001	−0.141	−0.103
Precipitation [mm]	0.914	−0.089	0.017	<0.0001	−0.123	−0.056
Season—summer	1.839	0.609	0.053	<0.0001	0.506	0.713
Season—spring	2.309	0.837	0.032	<0.0001	0.775	0.899
Season—winter	1.698	0.53	0.031	<0.0001	0.468	0.591

**Table 6 medicina-57-01216-t006:** Types of stroke in each moon phase.

Moon Phase	Hemorrhagic Stroke *N* = 10	Subarachnoid Hemorrhage *N* = 28	LACI *N* = 51	PACI *N* = 212	POCI *N* = 91	TACI *N* = 10
New moon	1 (10%)	6 (21.43%)	12 (23.5%)	54 (25.5%)	21 (23%)	2 (20%)
1st quarter moon	4 (40%)	6 (21.43%)	21 (41.2%)	50 (23.6%)	31 (34%)	1 (10%)
Full moon	2 (20%)	9 (32.14%)	8 (15.7%)	52 (24.5%)	17 (19%)	0
3rd quarter moon	3 (30%)	7 (25.0%)	10 (19.6%)	56 (26.4%)	22 (24%)	7 (70%)
p	0.7011	0.5697	0.137	1.000	0.524	0.011

LACI- lacunar cerebral ischemia, PACI- partial anterior circulation infarcts, POCI- posterior circulation infarct, TACI- total anterior circulation infarct.

## Data Availability

All data generated or analysed during this study are included in this published article (and its Appendix A).

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
