# Peer review of "The Influence of Selected Meteorological Factors on the Prevalence and Course of Stroke"

_medicina, 2021, doi:10.3390/medicina57111216_

Round 1
Reviewer 1 Report
This is an interesting article by Katarzyna Zaręba et al. The homeostasis of human body can be affected by weather changes and parameters and as a result haemodynamic parameters may be affected significantly. Comments on the functional outcome need to be supported by the analysis of covariates that can have a significant impact on the recovery like the administration of iv-thrombolysis and the rate of complications.
- Abstract: The following 2 sentences appear for the first time in the conclusions.
‘A combination of the following meteorological parameters: lower mean temperature and low sunshine, high humidity and high wind speed all increase the risk of stroke in the winter. High humidity combined with high precipitation, low wind speed and low sunshine in the autumn period are associated with the lowest stroke incidence risk.’
The question is raised how they appear in the conclusion without anything mentioned in the results section. Please move to the results section.
- Whole manuscript: There is room for improvement in the use of English language and specific stroke related terms. Examples: In lines 140-141, presented with ischaemic heart disease. In line 145, we do not use the term ‘cardiogenic’ in stroke medicine, at least not widely. Maybe you want to consider the term cardioembolic unless you include hypoperfusion due to low ejection fraction.
- Introduction: ‘The main objective of this study was to evaluate the impact of selected meteorological factors and their variability’: I am not sure the impact of variability is assessed. I suggest removing the term variability.
- Material and methods: ‘Additionally, stroke risk factors diagnosed during hospitalization were included in the assessment’: Were all investigations completed within the 14 days of admission for all patients? If not were post discharge investigations included? If they were then please re-word this phrase.
- Material and methods: How long was the prolonged arrhythmia monitoring? Please define. Did patients have a cardiac ultrasound as well as part of their investigations? Please describe if this was routine practice or The
- Materials and methods: Please split the first paragraph into smaller ones. This will facilitate the flow in reading the manuscript.
- Materials and methods: Regarding the prognosis there are a few important factors that are not taken into consideration by the authors such as any acute recanalization treatment in ischaemic stroke (e.g. intravenous thrombolysis) and the number, frequency and severity of complications.
- Materials and methods: Only patients hospitalized <14 days were included in the study and mRS scale was assessed on discharge date. This is a premature date for the assessment of the functional outcome and it is not unlikely that many of the patients with an mRS of >2 could have shifted to lower mRS scores in 3 months.
- Tables: In the NIHSS and mRS I suggest you remove the mean and use only the median. Values with decimals do not exist on either of these scales and the mean number are confusing.
- Results: The sentence ‘Older individuals were significantly more often burdened with cardiovascular disease and cardiogenic stroke’ needs rephrasing detailing which cardiovascular conditions were more frequent in the elderly, as there were a few of them that had the same prevalence as in the younger population (e.g. previous myocardial infarction, previous stroke etc.).
- Results: It is not clear to me what the authors want to say with their sentence ‘As per ASCOD classification, the most frequently observed stroke type in particular months was cardiogenic stroke and the type of stroke classified under “other” types (25.8% – 41.7% and 42.3% – 159 4%, respectively).’ Could you please clarify and possibly reword?
- Results: ‘Atrial fibrillation (HR 1.94 p=0.047) was an independent risk factor for severe neurological status (NIHSS 13–42) among the studied clinical and meteorological parameters.’ Did the analysis include ischemic and hemorrhagic strokes together? AF is a risk factor for ischaemic strokes only. If not, I would encourage the authors to do a sensitivity analysis for their ischaemic strokes only (the number of the ICHs is relatively small).
- Results: ‘None of the meteorological parameters proved to be an independent factor which influenced the neurological status as per the NIHSS.’ The term ‘influenced’ suggests a causality rather than a statistically significant correlation only which is a more appropriate approach.
- Conclusions: Conclusions about functional outcomes are significantly limited by the fact that a lot of factors that can have an effect in the functional outcome have not been taken into consideration (please see comment about methods above).
- Supplement, table 1: Why do the authors not use the ANOVA test to compare all 3 groups?
Author Response
Dear Reviewer,
we were happy to read the Reviewer’s and Editor's comments which we do appreciate. We have done our best to make sure that the paper itself, the changes introduced as per the Reviewer’s comments, and our responses meet your expectations.
Below we refer to each of the Reviewer’s comments and suggestions. The changes introduced to the text are highlighted in red.
Q & A:
Abstract: The following 2 sentences appear for the first time in the conclusions.
A combination of the following meteorological parameters: lower mean temperature and low sunshine, high humidity and high wind speed all increase the risk of stroke in the winter. High humidity combined with high precipitation, low wind speed and low sunshine in the autumn period are associated with the lowest stroke incidence risk.’
The question is raised how they appear in the conclusion without anything mentioned in the results section. Please move to the results section.
A: Thank you for this remark. We introduced a missing data to the Results section of the Abstract.
Whole manuscript: There is room for improvement in the use of English language and specific stroke related terms. Examples: In lines 140-141, presented with ischaemic heart disease. In line 145, we do not use the term ‘cardiogenic’ in stroke medicine, at least not widely. Maybe you want to consider the term cardioembolic unless you include hypoperfusion due to low ejection fraction.
A: we corrected all mentioned by Reviewer sentences including the „ischaemic” (though we think that both: ischemic” and ischaemic are corrected). We introduced cardioembolic instead of cardiogenic.
Introduction: ‘The main objective of this study was to evaluate the impact of selected meteorological factors and their variability’: I am not sure the impact of variability is assessed. I suggest removing the term variability.
A: The term variability was deleted.
Material and methods: ‘Additionally, stroke risk factors diagnosed during hospitalization were included in the assessment’: Were all investigations completed within the 14 days of admission for all patients? If not were post discharge investigations included? If they were then please re-word this phrase.
A: Thank you for this remark. We changed the description this issue in the section Methods. Not all planned diagnostic procedures and/or results were available during hospitalization.
Material and methods: How long was the prolonged arrhythmia monitoring? Please define. Did patients have a cardiac ultrasound as well as part of their investigations? Please describe if this was routine practice.
A: We introduced the information about the HR monitoring and TTE (TEE) in the section Methods.
Materials and methods: Please split the first paragraph into smaller ones. This will facilitate the flow in reading the manuscript.
A: The first paragraph was splitted and reduced to improve it readability.
Materials and methods: Regarding the prognosis there are a few important factors that are not taken into consideration by the authors such as any acute recanalization treatment in ischaemic stroke (e.g. intravenous thrombolysis) and the number, frequency and severity of complications.
A: The missing information about rt-PA iv was introduced in sections Methods and Results.
Materials and methods: Only patients hospitalized <14 days were included in the study and mRS scale was assessed on discharge date. This is a premature date for the assessment of the functional outcome and it is not unlikely that many of the patients with an mRS of >2 could have shifted to lower mRS scores in 3 months.
A: We agree with Reviewer, but we weren't able to collect complete data after 14 days. We introduced the lack of longer follow up as limitation of the study.
Tables: In the NIHSS and mRS I suggest you remove the mean and use only the median. Values with decimals do not exist on either of these scales and the mean number are confusing.
A: The results of NIHSS and mRS was changed according the the Reviewer’s suggestion.
Results: The sentence ‘Older individuals were significantly more often burdened with cardiovascular disease and cardiogenic stroke’ needs rephrasing detailing which cardiovascular conditions were more frequent in the elderly, as there were a few of them that had the same prevalence as in the younger population (e.g. previous myocardial infarction, previous stroke etc.).
A: We listed the cardiovascular risks factor of stroke in proper paragraph (in Results)
Results: It is not clear to me what the authors want to say with their sentence ‘As per ASCOD classification, the most frequently observed stroke type in particular months was cardiogenic stroke and the type of stroke classified under “other” types (25.8% – 41.7% and 42.3% – 159 4%, respectively).’ Could you please clarify and possibly reword?
A: We deeply changed the proper paragraph making it more understandable.
Results: ‘Atrial fibrillation (HR 1.94 p=0.047) was an independent risk factor for severe neurological status (NIHSS 13–42) among the studied clinical and meteorological parameters.’ Did the analysis include ischemic and hemorrhagic strokes together? AF is a risk factor for ischaemic strokes only. If not, I would encourage the authors to do a sensitivity analysis for their ischaemic strokes only (the number of the ICHs is relatively small).
A: We corrected the sentence and supplement the results to present data of all patients and only ischemic stroke patients.
Results: ‘None of the meteorological parameters proved to be an independent factor which influenced the neurological status as per the NIHSS.’ The term ‘influenced’ suggests a causality rather than a statistically significant correlation only which is a more appropriate approach.
A: We changed the sentence according the Reviewer’s suggestion.
Conclusions: Conclusions about functional outcomes are significantly limited by the fact that a lot of factors that can have an effect in the functional outcome have not been taken into consideration (please see comment about methods above).
A: We agree with Reviewer, but at current stage of the study we cannot improve it. We described it as limitation of our study.
Supplement, table 1: Why do the authors not use the ANOVA test to compare all 3 groups?
A: We introduced p value in the Table 1 in Supplement. The ANOVA test confirmed the lack of the statistical significance in analysis the correlation between the neurological state and meteorological parameters.
Dear Reviewer,
we were happy to read the Reviewer’s and Editor's comments which we do appreciate. We have done our best to make sure that the paper itself, the changes introduced as per the Reviewer’s comments, and our responses meet your expectations.
Below we refer to each of the Reviewer’s comments and suggestions. The changes introduced to the text are highlighted in red.
Q & A:
Abstract: The following 2 sentences appear for the first time in the conclusions.
A combination of the following meteorological parameters: lower mean temperature and low sunshine, high humidity and high wind speed all increase the risk of stroke in the winter. High humidity combined with high precipitation, low wind speed and low sunshine in the autumn period are associated with the lowest stroke incidence risk.’
The question is raised how they appear in the conclusion without anything mentioned in the results section. Please move to the results section.
A: Thank you for this remark. We introduced a missing data to the Results section of the Abstract.
Whole manuscript: There is room for improvement in the use of English language and specific stroke related terms. Examples: In lines 140-141, presented with ischaemic heart disease. In line 145, we do not use the term ‘cardiogenic’ in stroke medicine, at least not widely. Maybe you want to consider the term cardioembolic unless you include hypoperfusion due to low ejection fraction.
A: we corrected all mentioned by Reviewer sentences including the „ischaemic” (though we think that both: ischemic” and ischaemic are corrected). We introduced cardioembolic instead of cardiogenic.
Introduction: ‘The main objective of this study was to evaluate the impact of selected meteorological factors and their variability’: I am not sure the impact of variability is assessed. I suggest removing the term variability.
A: The term variability was deleted.
Material and methods: ‘Additionally, stroke risk factors diagnosed during hospitalization were included in the assessment’: Were all investigations completed within the 14 days of admission for all patients? If not were post discharge investigations included? If they were then please re-word this phrase.
A: Thank you for this remark. We changed the description this issue in the section Methods. Not all planned diagnostic procedures and/or results were available during hospitalization.
Material and methods: How long was the prolonged arrhythmia monitoring? Please define. Did patients have a cardiac ultrasound as well as part of their investigations? Please describe if this was routine practice.
A: We introduced the information about the HR monitoring and TTE (TEE) in the section Methods.
Materials and methods: Please split the first paragraph into smaller ones. This will facilitate the flow in reading the manuscript.
A: The first paragraph was splitted and reduced to improve it readability.
Materials and methods: Regarding the prognosis there are a few important factors that are not taken into consideration by the authors such as any acute recanalization treatment in ischaemic stroke (e.g. intravenous thrombolysis) and the number, frequency and severity of complications.
A: The missing information about rt-PA iv was introduced in sections Methods and Results.
Materials and methods: Only patients hospitalized <14 days were included in the study and mRS scale was assessed on discharge date. This is a premature date for the assessment of the functional outcome and it is not unlikely that many of the patients with an mRS of >2 could have shifted to lower mRS scores in 3 months.
A: We agree with Reviewer, but we weren't able to collect complete data after 14 days. We introduced the lack of longer follow up as limitation of the study.
Tables: In the NIHSS and mRS I suggest you remove the mean and use only the median. Values with decimals do not exist on either of these scales and the mean number are confusing.
A: The results of NIHSS and mRS was changed according the the Reviewer’s suggestion.
Results: The sentence ‘Older individuals were significantly more often burdened with cardiovascular disease and cardiogenic stroke’ needs rephrasing detailing which cardiovascular conditions were more frequent in the elderly, as there were a few of them that had the same prevalence as in the younger population (e.g. previous myocardial infarction, previous stroke etc.).
A: We listed the cardiovascular risks factor of stroke in proper paragraph (in Results)
Results: It is not clear to me what the authors want to say with their sentence ‘As per ASCOD classification, the most frequently observed stroke type in particular months was cardiogenic stroke and the type of stroke classified under “other” types (25.8% – 41.7% and 42.3% – 159 4%, respectively).’ Could you please clarify and possibly reword?
A: We deeply changed the proper paragraph making it more understandable.
Results: ‘Atrial fibrillation (HR 1.94 p=0.047) was an independent risk factor for severe neurological status (NIHSS 13–42) among the studied clinical and meteorological parameters.’ Did the analysis include ischemic and hemorrhagic strokes together? AF is a risk factor for ischaemic strokes only. If not, I would encourage the authors to do a sensitivity analysis for their ischaemic strokes only (the number of the ICHs is relatively small).
A: We corrected the sentence and supplement the results to present data of all patients and only ischemic stroke patients.
Results: ‘None of the meteorological parameters proved to be an independent factor which influenced the neurological status as per the NIHSS.’ The term ‘influenced’ suggests a causality rather than a statistically significant correlation only which is a more appropriate approach.
A: We changed the sentence according the Reviewer’s suggestion.
Conclusions: Conclusions about functional outcomes are significantly limited by the fact that a lot of factors that can have an effect in the functional outcome have not been taken into consideration (please see comment about methods above).
A: We agree with Reviewer, but at current stage of the study we cannot improve it. We described it as limitation of our study.
Supplement, table 1: Why do the authors not use the ANOVA test to compare all 3 groups?
A: We introduced p value in the Table 1 in Supplement. The ANOVA test confirmed the lack of the statistical significance in analysis the correlation between the neurological state and meteorological parameters.
Dear Reviewer,
we were happy to read the Reviewer’s and Editor's comments which we do appreciate. We have done our best to make sure that the paper itself, the changes introduced as per the Reviewer’s comments, and our responses meet your expectations.
Below we refer to each of the Reviewer’s comments and suggestions. The changes introduced to the text are highlighted in red.
Q & A:
Abstract: The following 2 sentences appear for the first time in the conclusions.
A combination of the following meteorological parameters: lower mean temperature and low sunshine, high humidity and high wind speed all increase the risk of stroke in the winter. High humidity combined with high precipitation, low wind speed and low sunshine in the autumn period are associated with the lowest stroke incidence risk.’
The question is raised how they appear in the conclusion without anything mentioned in the results section. Please move to the results section.
A: Thank you for this remark. We introduced a missing data to the Results section of the Abstract.
Whole manuscript: There is room for improvement in the use of English language and specific stroke related terms. Examples: In lines 140-141, presented with ischaemic heart disease. In line 145, we do not use the term ‘cardiogenic’ in stroke medicine, at least not widely. Maybe you want to consider the term cardioembolic unless you include hypoperfusion due to low ejection fraction.
A: we corrected all mentioned by Reviewer sentences including the „ischaemic” (though we think that both: ischemic” and ischaemic are corrected). We introduced cardioembolic instead of cardiogenic.
Introduction: ‘The main objective of this study was to evaluate the impact of selected meteorological factors and their variability’: I am not sure the impact of variability is assessed. I suggest removing the term variability.
A: The term variability was deleted.
Material and methods: ‘Additionally, stroke risk factors diagnosed during hospitalization were included in the assessment’: Were all investigations completed within the 14 days of admission for all patients? If not were post discharge investigations included? If they were then please re-word this phrase.
A: Thank you for this remark. We changed the description this issue in the section Methods. Not all planned diagnostic procedures and/or results were available during hospitalization.
Material and methods: How long was the prolonged arrhythmia monitoring? Please define. Did patients have a cardiac ultrasound as well as part of their investigations? Please describe if this was routine practice.
A: We introduced the information about the HR monitoring and TTE (TEE) in the section Methods.
Materials and methods: Please split the first paragraph into smaller ones. This will facilitate the flow in reading the manuscript.
A: The first paragraph was splitted and reduced to improve it readability.
Materials and methods: Regarding the prognosis there are a few important factors that are not taken into consideration by the authors such as any acute recanalization treatment in ischaemic stroke (e.g. intravenous thrombolysis) and the number, frequency and severity of complications.
A: The missing information about rt-PA iv was introduced in sections Methods and Results.
Materials and methods: Only patients hospitalized <14 days were included in the study and mRS scale was assessed on discharge date. This is a premature date for the assessment of the functional outcome and it is not unlikely that many of the patients with an mRS of >2 could have shifted to lower mRS scores in 3 months.
A: We agree with Reviewer, but we weren't able to collect complete data after 14 days. We introduced the lack of longer follow up as limitation of the study.
Tables: In the NIHSS and mRS I suggest you remove the mean and use only the median. Values with decimals do not exist on either of these scales and the mean number are confusing.
A: The results of NIHSS and mRS was changed according the the Reviewer’s suggestion.
Results: The sentence ‘Older individuals were significantly more often burdened with cardiovascular disease and cardiogenic stroke’ needs rephrasing detailing which cardiovascular conditions were more frequent in the elderly, as there were a few of them that had the same prevalence as in the younger population (e.g. previous myocardial infarction, previous stroke etc.).
A: We listed the cardiovascular risks factor of stroke in proper paragraph (in Results)
Results: It is not clear to me what the authors want to say with their sentence ‘As per ASCOD classification, the most frequently observed stroke type in particular months was cardiogenic stroke and the type of stroke classified under “other” types (25.8% – 41.7% and 42.3% – 159 4%, respectively).’ Could you please clarify and possibly reword?
A: We deeply changed the proper paragraph making it more understandable.
Results: ‘Atrial fibrillation (HR 1.94 p=0.047) was an independent risk factor for severe neurological status (NIHSS 13–42) among the studied clinical and meteorological parameters.’ Did the analysis include ischemic and hemorrhagic strokes together? AF is a risk factor for ischaemic strokes only. If not, I would encourage the authors to do a sensitivity analysis for their ischaemic strokes only (the number of the ICHs is relatively small).
A: We corrected the sentence and supplement the results to present data of all patients and only ischemic stroke patients.
Results: ‘None of the meteorological parameters proved to be an independent factor which influenced the neurological status as per the NIHSS.’ The term ‘influenced’ suggests a causality rather than a statistically significant correlation only which is a more appropriate approach.
A: We changed the sentence according the Reviewer’s suggestion.
Conclusions: Conclusions about functional outcomes are significantly limited by the fact that a lot of factors that can have an effect in the functional outcome have not been taken into consideration (please see comment about methods above).
A: We agree with Reviewer, but at current stage of the study we cannot improve it. We described it as limitation of our study.
Supplement, table 1: Why do the authors not use the ANOVA test to compare all 3 groups?
A: We introduced p value in the Table 1 in Supplement. The ANOVA test confirmed the lack of the statistical significance in analysis the correlation between the neurological state and meteorological parameters.
Dear Reviewer,
we were happy to read the Reviewer’s and Editor's comments which we do appreciate. We have done our best to make sure that the paper itself, the changes introduced as per the Reviewer’s comments, and our responses meet your expectations.
Below we refer to each of the Reviewer’s comments and suggestions. The changes introduced to the text are highlighted in red.
Q & A:
Abstract: The following 2 sentences appear for the first time in the conclusions.
A combination of the following meteorological parameters: lower mean temperature and low sunshine, high humidity and high wind speed all increase the risk of stroke in the winter. High humidity combined with high precipitation, low wind speed and low sunshine in the autumn period are associated with the lowest stroke incidence risk.’
The question is raised how they appear in the conclusion without anything mentioned in the results section. Please move to the results section.
A: Thank you for this remark. We introduced a missing data to the Results section of the Abstract.
Whole manuscript: There is room for improvement in the use of English language and specific stroke related terms. Examples: In lines 140-141, presented with ischaemic heart disease. In line 145, we do not use the term ‘cardiogenic’ in stroke medicine, at least not widely. Maybe you want to consider the term cardioembolic unless you include hypoperfusion due to low ejection fraction.
A: we corrected all mentioned by Reviewer sentences including the „ischaemic” (though we think that both: ischemic” and ischaemic are corrected). We introduced cardioembolic instead of cardiogenic.
Introduction: ‘The main objective of this study was to evaluate the impact of selected meteorological factors and their variability’: I am not sure the impact of variability is assessed. I suggest removing the term variability.
A: The term variability was deleted.
Material and methods: ‘Additionally, stroke risk factors diagnosed during hospitalization were included in the assessment’: Were all investigations completed within the 14 days of admission for all patients? If not were post discharge investigations included? If they were then please re-word this phrase.
A: Thank you for this remark. We changed the description this issue in the section Methods. Not all planned diagnostic procedures and/or results were available during hospitalization.
Material and methods: How long was the prolonged arrhythmia monitoring? Please define. Did patients have a cardiac ultrasound as well as part of their investigations? Please describe if this was routine practice.
A: We introduced the information about the HR monitoring and TTE (TEE) in the section Methods.
Materials and methods: Please split the first paragraph into smaller ones. This will facilitate the flow in reading the manuscript.
A: The first paragraph was splitted and reduced to improve it readability.
Materials and methods: Regarding the prognosis there are a few important factors that are not taken into consideration by the authors such as any acute recanalization treatment in ischaemic stroke (e.g. intravenous thrombolysis) and the number, frequency and severity of complications.
A: The missing information about rt-PA iv was introduced in sections Methods and Results.
Materials and methods: Only patients hospitalized <14 days were included in the study and mRS scale was assessed on discharge date. This is a premature date for the assessment of the functional outcome and it is not unlikely that many of the patients with an mRS of >2 could have shifted to lower mRS scores in 3 months.
A: We agree with Reviewer, but we weren't able to collect complete data after 14 days. We introduced the lack of longer follow up as limitation of the study.
Tables: In the NIHSS and mRS I suggest you remove the mean and use only the median. Values with decimals do not exist on either of these scales and the mean number are confusing.
A: The results of NIHSS and mRS was changed according the the Reviewer’s suggestion.
Results: The sentence ‘Older individuals were significantly more often burdened with cardiovascular disease and cardiogenic stroke’ needs rephrasing detailing which cardiovascular conditions were more frequent in the elderly, as there were a few of them that had the same prevalence as in the younger population (e.g. previous myocardial infarction, previous stroke etc.).
A: We listed the cardiovascular risks factor of stroke in proper paragraph (in Results)
Results: It is not clear to me what the authors want to say with their sentence ‘As per ASCOD classification, the most frequently observed stroke type in particular months was cardiogenic stroke and the type of stroke classified under “other” types (25.8% – 41.7% and 42.3% – 159 4%, respectively).’ Could you please clarify and possibly reword?
A: We deeply changed the proper paragraph making it more understandable.
Results: ‘Atrial fibrillation (HR 1.94 p=0.047) was an independent risk factor for severe neurological status (NIHSS 13–42) among the studied clinical and meteorological parameters.’ Did the analysis include ischemic and hemorrhagic strokes together? AF is a risk factor for ischaemic strokes only. If not, I would encourage the authors to do a sensitivity analysis for their ischaemic strokes only (the number of the ICHs is relatively small).
A: We corrected the sentence and supplement the results to present data of all patients and only ischemic stroke patients.
Results: ‘None of the meteorological parameters proved to be an independent factor which influenced the neurological status as per the NIHSS.’ The term ‘influenced’ suggests a causality rather than a statistically significant correlation only which is a more appropriate approach.
A: We changed the sentence according the Reviewer’s suggestion.
Conclusions: Conclusions about functional outcomes are significantly limited by the fact that a lot of factors that can have an effect in the functional outcome have not been taken into consideration (please see comment about methods above).
A: We agree with Reviewer, but at current stage of the study we cannot improve it. We described it as limitation of our study.
Supplement, table 1: Why do the authors not use the ANOVA test to compare all 3 groups?
A: We introduced p value in the Table 1 in Supplement. The ANOVA test confirmed the lack of the statistical significance in analysis the correlation between the neurological state and meteorological parameters.
Reviewer 2 Report
Dear manuscript authors,
The content of the manuscript presented is quite interesting, the type of research is retrospective, which includes the examination of the prevalence of stroke and its course under one or another meteorological condition. It is unfortunate only that the data were evaluated in one city, one hospital, and over a one-year period (only 402 cases in total), which specializes in treating stroke patients. The relatively small number of cases in this study is a major drawback of this article. According to this type of study, at least 5 years of data should be analyzed to obtain stable and meaningful results. The functional condition of the subjects assessed by the authors before discharge from the hospital and associated with certain meteorological parameters after illness is quite questionable. I think it depends more directly on other logistical, clinical, treatment, and other health parameters than on meteorological parameters that may or may not be related, but indirectly.
When evaluating a manuscript summary, it has all the sections needed for summaries. In the methodology section, the methods of statistical analysis used in the study could be briefly presented, the numbers of the subjects were taken from the results section. The odds ratios and other significant features obtained by Poisson regression could also be presented in the results section.
The introductory part of the manuscript is written quite well, the reader is introduced to the problem analyzed and evaluated in the article.
The methodological part of the manuscript has all the subsections specific to the methodological part. When describing stroke patients, they should be described in detail, indicating which stroke cases with which stroke types and ICD-10 codes were included in the study methodology. In general, this would require a more detailed description of the study design, what the inclusion and exclusion criteria were, and how many unclear cases were excluded from the study. Other sociodemographic and clinical factors that have been analyzed and evaluated, including arterial hypertension, diabetes, lipid metabolism disorder, atrial fibrillation, previous stroke, ischemic heart disease, former acute coronary syndrome, and so on, should be described in detail. (either based on medical records or anamnestic data) what methodologies were used to assess them? It has not been explained why only strokes that have been hospitalized for at least 14 days were included in the study. A more detailed description of the localization of the stroke according to the OSCP classification and the ASCOD scale should be provided or the source in which they are assessed should be indicated. When estimating meteorological factors, the wind speed is expressed in km/h (line 99), but the results are expressed in m/s (should be harmonized). A more detailed description of the first and third quarters of the month would also be needed. Interquartile differences in meteorological factors should also be reported in the statistical analysis section. Chi-square statistics I think can be used to compare both parametric and non-parametric data. When evaluating the results of logistic regression, it is unclear whether the data were adjusted in the model or not. The level of significance when comparing the data is not provided. Did the work not use z statistics to compare paired variables. It is not clear what features were calculated with the R environment.
The results are presented quite correctly in the results section, in tables, and in figures. There are some inaccuracies in the totals in the tables (Table 6, row 223). It is unclear why 2 age groups up to 65 and> 65 years were selected (Table 1), whereas the median was 72 years and the data would be analyzed in 2 neighborhood groups. In Figure 1, it would be better and more scientific to present the data as a percentage than in absolute numbers. Figure 2 is quite obvious, complicated, and the axis descriptions are partially correct. Table 3 should also provide 95% confidence intervals and here I do not think RR but OR must be provided and what statistics were used to obtain that data. It is unclear whether the data in this table were age-controlled because women were significantly older. What variables were still included in the model and whether there was a final logistic regression model to assess the functional status of the cases. Interesting data are presented in Table 4, where the mean blood pressure was higher at lower wind speeds. Table 5 presents Puasson regression data but evaluates the odds ratio when relative risk should be assessed (explanation needed).
In the discussion part, the authors of the manuscript present their obtained data in some detail and evaluate it with the data received by other authors or studies. This section provides a fairly good insight into the data, pointing out some of the pathogenetic mechanisms of certain significantly related variables. It also outlines the limitations of the study. I do not think that the retrospective type of research, by its very nature, imposes any limitations on research. Other lifestyle factors could be among the limitations, i.g. physical activity, nutrition type, harmful behavioral factors (alcohol and smoking), stress level and other psychosocial work environmental factors, social factors (education, job type), work and living environment, as well as some logistical and managerial aspects of the case that were not covered in this study. The strengths of the study should also be presented.
The findings could be more general as presented in the summary.
The names of the tables and graphs provided could be more accurate with an explanation of the abbreviations used.
The literature sources used to meet the requirements for literature sources, but should be numbered.
Author Response
Dear Reviewer,
we were happy to read the Reviewer’s and Editor's comments which we do appreciate. We have done our best to make sure that the paper itself, the changes introduced as per the Reviewer’s comments, and our responses meet your expectations.
Below we refer to each of the Reviewer’s comments and suggestions. The changes introduced to the text are highlighted in red.
Q & A:
The content of the manuscript presented is quite interesting, the type of research is retrospective, which includes the examination of the prevalence of stroke and its course under one or another meteorological condition. It is unfortunate only that the data were evaluated in one city, one hospital, and over a one-year period (only 402 cases in total), which specializes in treating stroke patients. The relatively small number of cases in this study is a major drawback of this article. According to this type of study, at least 5 years of data should be analyzed to obtain stable and meaningful results. The functional condition of the subjects assessed by the authors before discharge from the hospital and associated with certain meteorological parameters after illness is quite questionable. I think it depends more directly on other logistical, clinical, treatment, and other health parameters than on meteorological parameters that may or may not be related, but indirectly.
When evaluating a manuscript summary, it has all the sections needed for summaries. In the methodology section, the methods of statistical analysis used in the study could be briefly presented, the numbers of the subjects were taken from the results section. The odds ratios and other significant features obtained by Poisson regression could also be presented in the results section.
A: We introduced the missing results in section Results in Abstract.
The introductory part of the manuscript is written quite well, the reader is introduced to the problem analyzed and evaluated in the article.
A: Thank you for above commentary.
The methodological part of the manuscript has all the subsections specific to the methodological part. When describing stroke patients, they should be described in detail, indicating which stroke cases with which stroke types and ICD-10 codes were included in the study methodology. In general, this would require a more detailed description of the study design, what the inclusion and exclusion criteria were, and how many unclear cases were excluded from the study. Other sociodemographic and clinical factors that have been analyzed and evaluated, including arterial hypertension, diabetes, lipid metabolism disorder, atrial fibrillation, previous stroke, ischemic heart disease, former acute coronary syndrome, and so on, should be described in detail. (either based on medical records or anamnestic data) what methodologies were used to assess them? It has not been explained why only strokes that have been hospitalized for at least 14 days were included in the study. A more detailed description of the localization of the stroke according to the OSCP classification and the ASCOD scale should be provided or the source in which they are assessed should be indicated. When estimating meteorological factors, the wind speed is expressed in km/h (line 99), but the results are expressed in m/s (should be harmonized). A more detailed description of the first and third quarters of the month would also be needed. Interquartile differences in meteorological factors should also be reported in the statistical analysis section. Chi-square statistics I think can be used to compare both parametric and non-parametric data. When evaluating the results of logistic regression, it is unclear whether the data were adjusted in the model or not. The level of significance when comparing the data is not provided. Did the work not use z statistics to compare paired variables. It is not clear what features were calculated with the R environment.
A: We introduced many changes in our manuscript according to Reviewer’s suggestions. We introduced the ICD-10 codes, inclusion/exclusion criteria, and the definitions of comorbidities. We described deeply the types of strokes in our patients according the OSCP and ASCOD classifications. The data concerning the wind speed was unified. We explained the reason of including only short hospitalisations (10-14 days)- we wanted to have the complete data and similar stroke time when analyzing the functional state of patients on discharge day.
The results are presented quite correctly in the results section, in tables, and in figures. There are some inaccuracies in the totals in the tables (Table 6, row 223). It is unclear why 2 age groups up to 65 and> 65 years were selected (Table 1), whereas the median was 72 years and the data would be analyzed in 2 neighborhood groups. In Figure 1, it would be better and more scientific to present the data as a percentage than in absolute numbers. Figure 2 is quite obvious, complicated, and the axis descriptions are partially correct. Table 3 should also provide 95% confidence intervals and here I do not think RR but OR must be provided and what statistics were used to obtain that data. It is unclear whether the data in this table were age-controlled because women were significantly older. What variables were still included in the model and whether there was a final logistic regression model to assess the functional status of the cases. Interesting data are presented in Table 4, where the mean blood pressure was higher at lower wind speeds. Table 5 presents Puasson regression data but evaluates the odds ratio when relative risk should be assessed (explanation needed).
A: We modified the Tables 3,5, 6 and Figures 1, 2 according the Rewiever’s suggesions. We supplemented the section Results according to Rewiever’s remark (parameters potentially influence on mRS analysed in regression analysis were listed). We assumed that age-related risk factors for stroke (mainly atherosclerosis) might be relevant for the interaction between meteorological parameters and stroke, we chose the age limit of 65. According to the Reviewers’s suggesion concerning Table 3- we only have a limited data. The statistical program does not provide a confidence interval for the odds ratio. In the first version of article: RR was introduced instead of OR- the mistake was corrected.
In the discussion part, the authors of the manuscript present their obtained data in some detail and evaluate it with the data received by other authors or studies. This section provides a fairly good insight into the data, pointing out some of the pathogenetic mechanisms of certain significantly related variables. It also outlines the limitations of the study. I do not think that the retrospective type of research, by its very nature, imposes any limitations on research. Other lifestyle factors could be among the limitations, i.g. physical activity, nutrition type, harmful behavioral factors (alcohol and smoking), stress level and other psychosocial work environmental factors, social factors (education, job type), work and living environment, as well as some logistical and managerial aspects of the case that were not covered in this study. The strengths of the study should also be presented.
A: We modified Limitations and introduced the Strengths of our study.
The findings could be more general as presented in the summary.
A: We modified the conclusions making them more general.
The names of the tables and graphs provided could be more accurate with an explanation of the abbreviations used.
A: We change the titles of tables/graphs and we explained the abbreviations (in 3 tables, 1 graphs).
The literature sources used to meet the requirements for literature sources, but should be numbered.
A: We renumbered the references.
Round 2
Reviewer 1 Report
I think that the manuscript has been sufficiently improved.